# Physical Activity and Mental Health After COVID-19 Recovery: Age and Sex Differences

**DOI:** 10.3390/life15040531

**Published:** 2025-03-24

**Authors:** Miloš Stamenković, Saša Pantelić, Saša Bubanj, Emilija Petković, Bojan Bjelica, Nikola Aksović, Lazar Toskić, Constantin Sufaru, Gabriel-Stănică Lupu, Daniel-Lucian Dobreci, Tatiana Dobrescu, Mihai-Adrian Sava

**Affiliations:** 1Faculty of Sport and Physical Education, University of Niš, 18000 Niš, Serbia; kineziologija92@gmail.com (M.S.); spantelic2002@yahoo.com (S.P.); petkovicemilija@yahoo.com (E.P.); 2Faculty of Physical Education and Sports, University of East Sarajevo, 71126 Lukavica, Bosnia and Herzegovina; vipbjelica@gmail.com; 3Faculty of Sport and Physical Education, University of Priština-Kosovska Mitrovica, 38218 Leposavić, Serbia; kokir87np@gmail.com (N.A.); lazar.toskic@pr.ac.rs (L.T.); 4Faculty of Sport, University “Union–Nikola Tesla”, 11070 Belgrade, Serbia; 5Faculty of Movement, Sports and Health Sciences, “Vasile Alecsandri” University of Bacău, 600115 Bacău, Romania; sufaruconstantin@ub.ro (C.S.); gabi.lupu@ub.ro (G.-S.L.); dobreci.lucian@ub.ro (D.-L.D.); sava.adrian@ub.ro (M.-A.S.)

**Keywords:** physical activity, depression, anxiety, stress, COVID-19 recovery, differences, influence

## Abstract

(1) Background: The relationship between physical activity (PA) and mental health remains a topic of significant interest, particularly in populations recovering from COVID-19. This study aimed to examine the differences in PA levels and mental health parameters (anxiety, depression, and stress) across sex and age groups following COVID-19 recovery; (2) Methods: The sample of participants (*n* = 288) was selected using a random sampling method and consisted of participants of different ages and both sexes. Participants’ self-reported PA was assessed using the International Physical Activity Questionnaire (IPAQ), while the Depression, Anxiety, and Stress Scale (DASS) was used to measure negative emotional states. The influence of COVID-19 recovery on PA and mental health (depression, anxiety, stress) was analyzed using MANOVA and follow-up univariate tests by age and sex. Statistical significance was set at *p* < 0.05, with SPSS (v23.0) used for data analysis; (3) Results: A statistically significant difference was observed between the initial and final measurements in PA levels (Wilk’s Lambda = 0.95; *p* = 0.000) and mental health (Wilk’s Lambda = 0.96; *p* = 0.000) with older individuals—particularly women—demonstrating improved PA levels and better mental health outcomes. In contrast, younger women exhibited a decline in both domains, with increased anxiety, depression, and stress symptoms. While younger men showed increased PA, their mental health parameters were slightly elevated, though still within normal ranges; (4) Conclusions: Our findings suggest that older individuals were better adapted to post-COVID challenges due to maturity and life experience, while younger populations may require additional support. The study underscored the complexity of the PA–mental health relationship and highlighted the need for further research, particularly among younger individuals recovering from COVID-19.

## 1. Introduction

The Severe Acute Respiratory Syndrome Coronavirus 2 (COVID-19) pandemic has emerged as a significant new stressor, triggering various psychological reactions and mental health challenges [1]. Several studies have confirmed that the pandemic and the associated lockdown measures led to increased levels of anxiety and depression [2,3,4]. These findings highlight the negative impact of the COVID-19 pandemic on mental health.

In addition to the psychological consequences, research has also focused on how the pandemic affected physical activity (PA) levels. Regular PA plays a crucial role in maintaining and improving overall health and well-being, making it an essential lifestyle factor [5]. Unfortunately, restrictions and lockdown measures implemented during the pandemic significantly reduced PA levels while simultaneously contributing to a concerning rise in anxiety and depression [6].

A longitudinal study conducted by Najafipour and associates [7] revealed that quarantine measures led to increased hypersomnia, decreased PA levels, and heightened anxiety, particularly among young people and women. Similarly, research by Martínez-de-Quel and associates [8] found that the pandemic and lockdowns negatively impacted PA levels, sleep quality, and overall well-being among physically active individuals.

Mattioli and associates [9] also reported a decline in PA levels during quarantine, which resulted in a rise in sedentary behaviors [10]. Further supporting this, Stockwell and associates [10] analyzed changes in PA and sedentary behavior before and during the pandemic, concluding that COVID-19 restrictions significantly reduced PA levels while increasing sedentary time. These findings emphasize the widespread consequences of the pandemic on both mental and physical health, underlining the need for strategies to mitigate these adverse effects.

Most existing studies focused either on the general population’s PA levels during lockdowns or on the mental health consequences of the pandemic. Few have examined how these factors evolve post-recovery, and even fewer have compared sex and age-related differences in this process. It remains unclear whether individuals regain pre-pandemic PA levels after recovery and how changes in PA correlate with mental health outcomes. Moreover, while it is well-documented that women tend to experience higher anxiety and depression rates than men, the interaction between sex, age, and post-COVID-19 PA remains underexplored. Understanding these variations is essential for designing targeted interventions aimed at improving both physical and mental health outcomes.

The primary aim of this study was to investigate the influence of COVID-19 recovery on PA and mental health parameters across different age and sex groups. Specifically, the study aimed to determine whether there were significant differences in PA levels and mental health outcomes between the initial and final measurements.

We hypothesized that there will be a statistically significant difference in PA levels and mental health parameters between the initial and final measurements following COVID-19 recovery.

## 2. Materials and Methods

In this study, a longitudinal research model was applied. Appropriate procedures were implemented in accordance with the defined research aims, tasks, and hypothesis.

Previous studies have shown that SARS-CoV-2 infection not only causes physical symptoms but also leads to changes in mental health, such as increased anxiety, depression, and stress [11,12]. Additionally, it is well known that SARS-CoV-2 negatively impacts individuals’ functional status, resulting in reduced physical activity levels and cardiorespiratory fitness [13,14]. The disease remains highly complex from a virological perspective, as the mutational mechanisms of the virus are still not fully understood [15].

This means that each new mutation may lead to different symptom manifestations, potentially altering the immune response over time [16].

Despite growing immune resistance due to the ability of T memory cells to recognize the pathogen [17], the question remains as to whether radical changes in the structure of SARS-CoV-2 will occur in the future. Understanding this interaction between the pathogen and the body’s immune response [18] helps provide context for the two measurement time points in our study. While we do not know whether some participants experienced post-COVID syndrome [19], we observed that physical activity levels varied among all participants, regardless of sex or age [20]. Moreover, the time interval between the initial and final measurements revealed that mental health parameters, such as anxiety, depression, and stress, were subject to change [21].

It is important to emphasize that, at no point, were participants given specific instructions by researchers regarding how much physical activity to engage in or how to increase its intensity.

### 2.1. Sample of Participants

The study was conducted in collaboration with the institutions of Leskovac Health Center and Leskovac General Hospital. Data on the participants, necessary for the research, were collected based on the approval of research requests by the Ethics Committees of these two healthcare institutions.

The sample was selected using a random sampling method and consisted of participants of different ages and both sexes from different locations of the Jablanica district who had recovered from COVID-19. Initially, the study aimed to include 320 participants (*n* = 320). However, the final number of participants was 288 (*n* = 288) (Figure 1). The research was conducted between February and May 2022. The initial measurement took place after participants had recovered from a COVID-19 infection caused by the Omicron variant, which was the most transmissible strain at the time. The final measurement was conducted three months later.

To participate in the study, specific criteria had to be met. Therefore, inclusion and exclusion criteria were established to determine eligibility.

Inclusion criteria:Participants of various ages and both sexes who did not experience severe clinical symptoms and were not on ventilators.Diagnosis of COVID-19.Home treatment.Hospitalization for up to seven days.A maximum of one month had passed since hospital discharge and self-isolation.No mental illnesses requiring specific medication.

Exclusion criteria:Participants who experienced severe clinical symptoms.Those who were on ventilators.Individuals without a confirmed COVID-19 diagnosis.Those who were not treated at home.Hospitalized for more than seven days.More than a month had passed since hospital discharge and self-isolation.Participants with mental illnesses requiring specific medication.

### 2.2. Ethical Considerations

Before testing began, participants were fully informed about the benefits and potential risks of testing and participation in the study. Each participant provided consent for voluntary participation before the study commenced. As obtaining signed consent from all participants in person was not feasible, some participants received consent forms via email as a Word document.

Thanks to the directors of the General Hospital Leskovac, the Health Center Leskovac, health stations, and local municipalities, the research team was able to obtain phone numbers, email addresses, and other relevant information for the study. Participants could withdraw from the study at any time.

The questionnaire was conducted via telephone interviews. Data were collected from participants twice, with a sufficient interval between the first and second IPAQ test to minimize any influence of initial responses on the retest results. Participants reported their physical activity over the previous seven days, detailing the number of days and duration spent on vigorous and moderate-intensity activities, as well as walking, across all four test domains [22]. All questionnaires were coded to facilitate the identification of participants from the initial and final testing and to compare their results over time. All personal data were securely protected and never misused. Participants were thoroughly informed about the research procedures and the necessity for a follow-up assessment three months after the initial testing.

The study was conducted following the Declaration of Helsinki and ethical guidelines, with approval from the Ethics Committees of the University of Niš, Health Center Leskovac, and General Hospital Leskovac.

### 2.3. Physical Activity Assessment

Participants’ self-reported PA was assessed using the International Physical Activity Questionnaire (IPAQ), a validated instrument for scientific research [22]. Key parameters analyzed included frequency, intensity, and duration of PA. The questionnaire is suitable for participants aged 15 to 69 and examines PA across eight domains: (1) PA at work; (2) PA related to transportation; (3) Leisure-time PA; (4) Household and gardening activities; (5) Walking; (6) Moderate-intensity PA; (7) High-intensity PA; (8) Total PA.

The longer version of the IPAQ provides a comprehensive assessment of PA levels in different settings, detailing the duration of activities in all eight domains and the number of days per week participants engaged in these activities. This extended version offers a more precise methodological assessment compared to the shorter version.

A metabolic equivalent (MET) score is calculated for each domain to better quantify and classify participants’ PA levels. The total weekly MET-minutes are summed to determine overall activity levels.

Results are presented in MET-minutes per week. To obtain these numerical values, participants recorded the total minutes per day engaged in activities and the number of days per week. These values were multiplied by MET coefficients that indicate activity intensity. MET coefficients were calculated for each type of activity. For example, various types of walking were assessed to determine an average MET value, and the same procedure was applied for moderate- and high-intensity activities (Table 1).

### 2.4. Mental Health Assessment

The Depression, Anxiety, and Stress Scale (DASS) [23] was used to measure negative emotional states. The DASS consists of 42 items rated on a four-point scale and is designed to improve the definition, understanding, and measurement of prevalent and clinically significant emotional states, such as depression, anxiety, and stress.

Each of the three DASS subscales contains 14 items, grouped into smaller subscales of 2–5 items with similar content: (1) Depression scale assesses dysphoria, hopelessness, devaluation of life, self-deprecation, lack of interest or engagement, anhedonia, and inertia; (2) Anxiety scale evaluates autonomic arousal, skeletal muscle effects, situational anxiety, and subjective experience of anxious affect; (3) Stress scale measures chronic nonspecific arousal, difficulty relaxing, nervous tension, irritability, overreactivity, and impatience.

Participants rated the degree to which they experienced each emotional state in the past week using a four-point severity/frequency scale. Scores for depression, anxiety, and stress were calculated by summing responses for the relevant items. The scale has demonstrated strong psychometric properties [24].

### 2.5. Statistical Procedures

Data processing employed the statistical program SPSS (v23.0, SPSS Inc., Chicago, IL, USA). To analyze the influence of COVID-19 recovery on PA and mental health parameters (depression, anxiety, and stress) across different age and sex groups, multivariate statistical analyses were conducted. Following a significant MANOVA result, univariate analyses were conducted to determine specific differences for each dependent variable across different age and sex groups. The level of significance (*p*) was set at *p* < 0.05 level to determine whether the observed differences were statistically significant.

## 3. Results

The study sample encompassed a diverse range of sociodemographic characteristics (Table 2).

Regarding education, the majority of participants had either a university degree (49.7%) or a high school diploma (45.5%), while a smaller proportion held a master’s or doctoral degree (3.8%) or had only completed primary school (1.0%). In terms of employment status, most participants were employed (91.7%), with smaller proportions being students (2.8%), unemployed individuals (1.0%), or retirees (3.5%). As for smoking habits, the majority of participants were non-smokers (71.2%), while 28.8% reported being smokers. When it comes to marital status, 74.7% of the participants were married, whereas 25.3% were single. Lastly, in terms of place of residence, the majority lived in urban areas (76.0%), while 24.0% resided in rural areas (Table 2).

Table 3 presents the results of a multivariate analysis of variance (MANOVA) examining changes in physical activity between initial and final measurements across different age and sex groups. The results indicate a statistically significant overall difference between groups (*p* = 0.000).

The univariate analysis of variance (ANOVA) revealed significant differences between initial and final measurements in the following variables: Transport-related PA (*p* = 0.006; ηp^2^ = 0.01); Leisure-time PA (*p* = 0.017; ηp^2^ = 0.01); Walking PA (*p* = 0.001; ηp^2^ = 0.01); Vigorous PA (*p* = 0.002; ηp^2^ = 0.01); and Total PA (*p* = 0.000; ηp^2^ = 0.02). However, no significant differences were observed in the following: Work-related PA (*p* = 0.426; ηp^2^ = 0.00); Household PA (*p* = 0.540; ηp^2^ = 0.00); and Moderate PA (*p* = 0.103; ηp^2^ = 0.00). Since the significance values for these variables exceed 0.05 (*p* > 0.05), the differences observed in these areas are not statistically meaningful.

Table 4 presents the results of the multivariate variance analysis (MANOVA) of mental health parameters among participants of different sexes between the initial and final measurements. The MANOVA results indicated a statistically significant difference between the groups (*p* = 0.000).

The univariate variance analysis (ANOVA) results further revealed significant intergroup differences in all mental health parameters between the initial and final measurements: depression (*p* = 0.000, ηp^2^ = 0.02), anxiety (*p* = 0.000, ηp^2^ = 0.03), and stress (*p* = 0.001, ηp^2^ = 0.02).

## 4. Discussion

The aim of this study was to examine whether there are differences between initial and final measurements in levels of PA and mental health parameters among men and women of different age groups who had recovered from COVID-19. The study results indicated a statistically significant difference at the multivariate level in both PA levels (*p* = 0.000) and mental health parameters (*p* = 0.000).

### 4.1. Differences in Physical Activity Levels Between Initial and Final Measurements

Regarding physical activity, detailed analysis of Table 3 shows that, at both initial and final measurements, younger and older men had higher MET values for overall physical activity compared to younger and older women. These findings align with previous research indicating that men tend to be more physically active than women [25,26,27].

One possible explanation for lower physical activity levels among women compared to men could be social status. In many countries, women’s societal roles include childcare and household management, which may limit opportunities for leisure-time physical activity [28].

Univariate analysis further revealed significant sex and age-related differences in specific physical activity domains: transport-related physical activity, walking, and vigorous physical activity.

Conversely, no statistically significant differences were observed in occupational physical activity, household physical activity, and moderate physical activity. These results were expected for occupational activity, as participants were part of a working-age population.

Interestingly, our study did not find significant differences in household physical activity and moderate-intensity physical activity between men and women of different age groups. Generally, men engage in activities requiring greater physiological effort [29,30,31]. Given that participants in our study had recovered from COVID-19, these results may reflect an adaptation to physical activity post-recovery. Work profiles of participants should also be considered when analyzing these findings.

Regarding household activity, previous research suggests that it is typically higher among women than men [32,33]. Our study found that younger women had higher MET values for household activities at both measurements compared to younger men. However, at the final measurement, older men exhibited higher MET values than older women. The variability in these results may stem from the broad range of activities included in household physical activity, as defined by the IPAQ questionnaire.

### 4.2. Differences in Mental Health Parameters Between Initial and Final Measurements

Our study confirmed significant sex and age-related differences in mental health parameters, including depression, anxiety, and stress (Table 4). The DASS questionnaire results indicate that younger and older women reported higher levels of depression, anxiety, and stress at both measurements compared to men. These findings are consistent with previous research showing that women experience higher levels of anxiety, depression, and stress than men [34,35,36].

Women’s increased susceptibility to anxiety may be partly due to biological factors such as hormonal fluctuations related to pregnancy and menstruation [34]. Additionally, depression prevalence among women is often linked to complex interactions between genetic, biochemical, hormonal, social, and psychological factors. Social and cultural expectations, lower social support, and exposure to abuse may also contribute to higher anxiety, depression, and stress levels among women.

Notably, stress levels fluctuated between initial and final measurements, reinforcing the dynamic nature of mental health. Among all mental health parameters, stress stood out, particularly among younger and older women. These results align with previous research showing that stress is more prevalent in women than men [37]. While the higher stress levels in women were expected, the reasons for elevated stress in men are less clear. From a psychological perspective, stress can be categorized as life stress (resulting from everyday life events) and occupational stress (related to work demands) [38]. Our findings suggest that occupational and daily life activities may contribute significantly to stress levels in men, regardless of their physical activity levels.

### 4.3. Age, Physical Activity, and Mental Health

Our research highlights that, in addition to gender, age plays a significant role in physical activity levels. A declining trend in physical activity was observed among older men and women compared to their younger counterparts. Although total physical activity at the final measurement exceeded 3000 MET-minutes, it was still lower than that of younger participants. Previous studies have also reported a decrease in physical activity with age [39,40]. However, this trend can be mitigated through properly structured physical activity programs that consider individual differences.

Interestingly, older women in our study demonstrated better results across all levels and domains of physical activity, as well as in mental health parameters, at the final measurement compared to the initial one.

Regarding mental health, older women showed lower levels of anxiety, depression, and stress at the final measurement compared to the initial one. While fluctuations in physical activity levels and mental health parameters were observed among younger participants of both genders, as well as among older men, this was not the case for older women.

In terms of physical activity, older women were primarily engaged in moderate-intensity activities, which made the greatest contribution to their total MET values at the final measurement. Based on these data, we suggest that the reduction in anxiety, depression, and stress symptoms observed at the final measurement was largely due to moderate physical activity. Previous research has shown that moderate physical activity can lead to a reduction in anxiety, depression, and stress symptoms [41,42]. Our conclusion is that moderate physical activity was the most beneficial for older women, which may explain the improvement in mental health indicators at the final measurement.

## 5. Conclusions

Our research results indicated a statistically significant difference between the initial and final measurements in levels and domains of PA, as well as in mental health parameters, among men and women of different ages after recovering from COVID-19. The findings also highlighted that sex continued to play a crucial role in PA.

Regarding mental health, women exhibited higher levels of anxiety, depression, and stress compared to men, regardless of age. A detailed analysis of individual results revealed that older women, compared to their initial measurements, showed improvements across all levels and domains of PA, along with lower anxiety, depression, and stress scores. In contrast, younger women experienced a deterioration in symptoms of depression, anxiety, and stress, as well as a decline in most domains and levels of PA.

Among both younger and older men, the final measurements indicated an overall increase in PA levels. However, younger men showed a slight increase in anxiety, depression, and stress values, though still within normal limits. These findings suggest that older men and women are better adapted and more rational due to their age, maturity, and life experience, whereas younger individuals—both men and women—appear to struggle more with adaptation.

Compared to older and younger men, as well as younger women, older women demonstrated the most favorable outcomes in all examined variables related to PA and mental health. These results underscore the complexity of the relationship between PA and mental health. Given the apparent variability among younger individuals recovering from COVID-19, further research is essential to better understand this intricate connection, particularly in younger populations.

## 6. Advantages and Shortcomings of This Study

From a methodological perspective, it is essential to highlight both the strengths and limitations of this study.

One of the key strengths of our research is that it did not focus solely on gender but also considered age differences. Our findings suggest that physical activity levels fluctuate with age, reinforcing the idea that such data can be valuable for designing and tailoring physical activity programs for individuals of different age groups, especially after recovering from COVID-19.

Another advantage is the ample sample size, which remained sufficiently large at both the initial and final measurements. Additionally, we used the long-form version of the IPAQ questionnaire, which contains more questions than the short version, allowing for a more detailed assessment of participants’ physical activity levels. Similarly, the long-form version of the DASS questionnaire was employed, enabling a more comprehensive analysis of mental health parameters.

Despite these strengths, it is also important to acknowledge the limitations of our study. Our research primarily relied on self-reported questionnaires and telephone interviews, which may introduce subjective bias. To achieve a more accurate assessment of actual physical activity levels, future studies should incorporate objective measurement tools based on biomechanical principles.

## Figures and Tables

**Figure 1 life-15-00531-f001:**
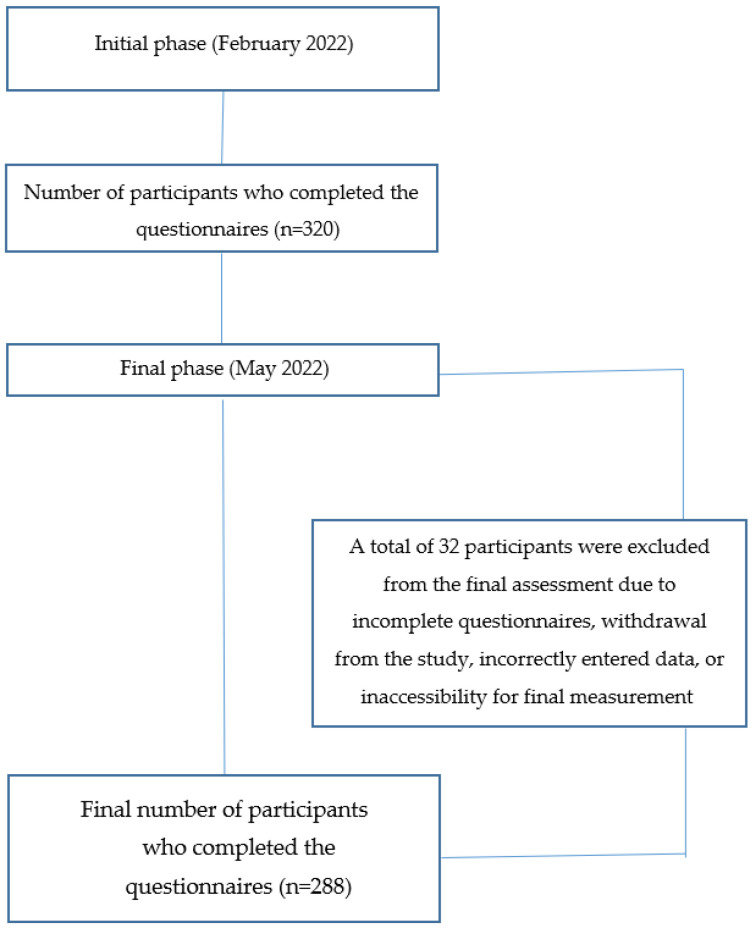
Illustration of the participant inclusion and exclusion process.

**Table 1 life-15-00531-t001:** Example of calculation of MET coefficients.

Walking MET-min/week = 3.3 × time spent in walking × number of days spent in walking.
Moderate physical activity MET-min/week = 4.0 × moderate physical activity in minutes × moderate physical activity in days.
High-intensity physical activity MET-min/week = 8.0 × high-intensity physical activity in minutes × high-intensity physical activity in days.
Total physical activity MET-min/week = sum for walking + for moderate physical activity + for high-intensity physical activity MET-min/week.

**Table 2 life-15-00531-t002:** Sociodemographic data of participants (*n* = 288).

Sociodemographic Characteristics				
Education	Primary School	High School	University Degree	Master’s/PhD
	3 (1.0%)	131 (45.5%)	143 (49.7%)	11 (3.8%)
Employment Status	Student	Employed	Unemployed	Retired
	8 (2.8%)	264 (91.7%)	3 (1.0%)	10 (3.5%)
Smoking Status	Smoker		Non-Smoker	
	83 (28.8%)		205 (71.2%)	
Marital Status	Married		Single	
	215 (74.7%)		73 (25.3%)	
Place of Residence	Rural		Urban	
	69 (24%)		219 (76%)	

**Table 3 life-15-00531-t003:** Multivariate and univariate analysis of physical activity among men and women of different ages between initial and final measurements.

	Variables	YM (I-F) (20–40 Years)	OM (I-F) (40–60 Years)	YW (I-F) (20–40 Years)	OW (I-F) (40–60 Years)	F	*p*	ηp^2^
PA is expressed in MET-minutes per week	Work-related PA	1503–1831	1417–1338	1351–1189	1178–1298	0.63	0.426	0.00
	Transport-related PA	478–545	431–521	463–458	348–419	7.75	0.006	0.01
	Household PA	1086–1081	1401–1538	1443–1449	1114–1465	0.37	0.540	0.00
	Leisure-time PA	1468–1239	1011–1136	1265–1130	746–859	5.71	0.017	0.01
	Walking PA	1286–1300	1029–1207	1229–1233	868–1010	10.96	0.001	0.01
	Moderate PA	1969–2186	2268–2506	2389–2081	1880–2284	2.66	0.103	0.00
	Vigorous PA	1279–1211	899–811	904–870	662–803	10.04	0.002	0.01
	Total PA	4535–4698	4141–4526	4523–4185	3411–4098	16.14	0.000	0.02
	Wilk’s Lambda = 0.95, F 3.70, df1 8, df2 567, *p* = 0.000							

Legend: Wilks’ Lambda—Wilks’ Lambda test statistic; F—Rao’s F approximation; df—Degrees of freedom; PA—Physical activity; *p*—Level of significance; ηp^2^—Partial eta squared (effect size); YM—Younger men; OM—Older men; YW—Younger women; OW—Older women; I—Initial measurement; F—Final measurement.

**Table 4 life-15-00531-t004:** Multivariate and univariate analysis of mental health parameters among participants of different sexes between initial and final measurements.

	Variables	YM (I-F) (20–40 Years)	OM (I-F) (40–60 Years)	YW (I-F) (20–40 Years)	OW (I-F) (40–60 Years)	F	*p*	ηp^2^
DASS is expressed as raw scores (0–42)	Depression	5.80–6.34	6.30–6.43	6.07–6.77	10.72–9.17	14.18	0.000	0.02
	Anxiety	6.60–7.14	7.15–6.57	7.66–8.80	11.81–9.74	22.21	0.000	0.03
	Stress	12.29–12.60	11.87–12.25	12.30–13.50	16.16–14.75	11.87	0.001	0.02
	Wilk’s Lambda = 0.96, F 7.43, df1 3, df2 572, *p* = 0.000							

Legend: Wilk’s Lambda—Wilks’ Lambda test, F—Rao’s F approximation, df—degrees of freedom, *p*—significance level, ηp^2^—partial eta squared test, YM—younger men, OM—older men, YW—younger women, OW—older women, I—initial, F—final measurement.

## Data Availability

The authors will provide the raw data supporting the conclusions of this article upon request.

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
