# Peer review of "Physical Activity and Mental Health After COVID-19 Recovery: Age and Sex Differences"

_life, 2025, doi:10.3390/life15040531_

Round 1
Reviewer 1 Report
Comments and Suggestions for Authors
This study examined differences in physical activity (PA) levels and mental health (i.e., anxiety, depression, and stress) across sex, and age groups follow COVID-19 pandemic. Results showed that older individuals had better PA and mental health levels at their second measurement.
Overall, this study lacks information in the methods and the importance of the timing of the measurements (initial vs. final phase) needs to be explained. Below are a few comments/questions related to the time points at which the measurements were taken:
What is the significance of taking both measurements in 2022 (2 years after the pandemic began)?
Was the initial measurement taken before participants had COVID, then the final measurement was taken after they had recovered from COVID? Or were both initial and final measurements taken after participants already had COVID? If so, what is the significance of these two time points? Please consider providing justification for these time points.
Consider adding age or age range as a descriptive statistic instead of "various ages."
Figure 1: How many participated in the initial phase?
Figure 1: For reasons that participants were missing data, include the numbers for each reason.
Ethical considerations section: Consider cutting this section down to a sentence or two.
Discussion: consider deleting statistics from the discussion section.
Author Response
Dear Reviewer,
Thank you very much for taking the time to review our manuscript. Your feedback has been invaluable and has significantly contributed to its improvement.
Please find our detailed responses, along with the corresponding revisions and corrections highlighted in the attached documents.
Thank you once again for your valuable input.
Kind regards,
The Authors

Reviewer 2 Report
Comments and Suggestions for Authors
-
It is necessary to explain the method of random selection.
-
In the "Ethical Considerations" section, the authors state, "...The questionnaire was conducted via telephone interviews…" – it is recommended to reference a source publication that describes the data collection process in detail (e.g., Reliability of the Serbian version of the International Physical Activity Questionnaire for older adults. Zoran Milanović, Saša Pantelić, Nebojša Trajković, Bojan Jorgić, Goran Sporiš & Milovan Bratić. Pages 581-587, 2014).
-
Instead of citing publication No. 11 from 2003, it is suggested to cite your 2014 publication (Reliability of the Serbian version of the International Physical Activity Questionnaire for older adults. Zoran Milanović, Saša Pantelić, Nebojša Trajković, Bojan Jorgić, Goran Sporiš & Milovan Bratić. Pages 581-587, 2014), where you provide your own version of the IPAQ.
-
The study sample does not specify the age and sex of the participants, which is crucial for the analyses. What do the abbreviations YM – Younger men; OM – Older men; YW – Younger women; OW – Older women mean exactly?
-
There is no information on how the MET index was calculated.
-
The authors state, "...The aim of this study was to determine whether there were differences between the initial and final measurements of physical activity levels...". What do the initial and final measurements refer to? When were the initial and final assessments conducted? Only one result is presented. Consequently, in the discussion and in the section on strengths and limitations, the authors claim that "...the results indicate that PA levels change with age, which may be important for planning physical activity programs for individuals recovering from COVID-19...". Additionally, the age of the participants and how it influenced the results remain unclear.
-
The entire discussion section:
"...Univariate analysis further revealed significant sex and age-related differences in specific PA domains: transport-related PA (p = .006, ηp2 = 0.01), leisure-time PA (p = .017, ηp2 = 0.01), walking (p = .001, ηp2 = 0.01), and vigorous PA (p = .002, ηp2 = 0.01). The partial eta squared test results suggest small but meaningful differences in these domains. Conversely, no statistically significant differences were observed in occupational PA (p = .426, ηp2 = 0.00), household PA (p = .540, ηp2 = 0.00), and moderate PA (p = .103, ηp2 = 0.00)."
is a repetition of the study results.
Author Response

(The authors gave the same response as above.)
